# Laboratory Selection of Trypanosomatid Pathogens for Drug Resistance

**DOI:** 10.3390/ph15020135

**Published:** 2022-01-24

**Authors:** Sabina Beilstein, Radhia El Phil, Suzanne Sherihan Sahraoui, Leonardo Scapozza, Marcel Kaiser, Pascal Mäser

**Affiliations:** 1Department Medical Parasitology and Infection Biology, Swiss Tropical and Public Health Institute, 4051 Basel, Switzerland; sabina.beilstein@swisstph.ch (S.B.); marcel.kaiser@swisstph.ch (M.K.); 2Swiss TPH, University of Basel, Petersplatz 1, 4002 Basel, Switzerland; 3School of Pharmaceutical Sciences, University of Geneva, 1205 Geneva, Switzerland; Radhia.ElPhil@unige.ch (R.E.P.); Suzanne.Sahraoui@unige.ch (S.S.S.); Leonardo.Scapozza@unige.ch (L.S.); 4Institute of Pharmaceutical Sciences of Western Switzerland, University of Geneva, 1211 Geneva, Switzerland

**Keywords:** *Trypanosoma brucei*, *Trypanosoma cruzi*, *Leishmania*, drug resistance, in vitro cultivation

## Abstract

The selection of parasites for drug resistance in the laboratory is an approach frequently used to investigate the mode of drug action, estimate the risk of emergence of drug resistance, or develop molecular markers for drug resistance. Here, we focused on the How rather than the Why of laboratory selection, discussing different experimental set-ups based on research examples with *Trypanosoma brucei*, *Trypanosoma cruzi*, and *Leishmania* spp. The trypanosomatids are particularly well-suited to illustrate different strategies of selecting for drug resistance, since it was with African trypanosomes that Paul Ehrlich performed such an experiment for the first time, more than a century ago. While breakthroughs in reverse genetics and genome editing have greatly facilitated the identification and validation of candidate resistance mutations in the trypanosomatids, the forward selection of drug-resistant mutants still relies on standard in vivo models and in vitro culture systems. Critical questions are: is selection for drug resistance performed in vivo or in vitro? With the mammalian or with the insect stages of the parasites? Under steady pressure or by sudden shock? Is a mutagen used? While there is no bona fide best approach, we think that a methodical consideration of these questions provides a helpful framework for selection of parasites for drug resistance in the laboratory.

## 1. The TriTryp Parasites

*Trypanosoma brucei*, *Trypanosoma cruzi* and *Leishmania* comprise the human-pathogenic species in the trypanosomatid family. They cause the neglected tropical diseases sleeping sickness, Chagas disease and leishmaniasis, which have an estimated total prevalence of over 10 million and impose a substantial burden on global health [1,2]. The insect vectors—tsetse flies, triatomine bugs and phlebotomine sandflies, respectively—transmit the parasites to the mammalian host during a blood meal. The parasites thus encounter very different environments in their transmission cycles. Furthermore, the three species are zoonotic and infect various mammals. *Leishmania* and *T. cruzi* are intracellular in the mammalian host and extracellular in the gut of the insect, whereas all life-cycle stages of *T. brucei* are extracellular. This renders *T. brucei* easier to cultivate than *T. cruzi* or *Leishmania*. In addition, *T. brucei* is more readily amenable to reverse genetics than other trypanosomatids [3]. Table 1 compares the cellular and molecular characteristics of the TriTryp parasites *T. brucei, T. cruzi* and *L. donovani.* The drugs for treating the respective diseases are listed in Table 2, along with the in vitro sensitivity of the different life-cycle stages of the parasites. These drugs were developed by phenotypic, cell-based approaches rather than target-based. Consequently, for some of them, even though they have been used for decades, the mechanism of action is still not fully understood.

## 2. New Tools for Target Deconvolution

Knowing a drug’s target is of great importance for the development of more effective, better tolerated therapies and for the management of drug resistance. However, the genetic mapping of mutations conferring resistance, which in *Caenorhabditis elegans* was key to the identification of anthelmintic drug targets [15], is precluded in trypanosomatids as sexual recombination, if it occurs at all, is not obligate. This lack of forward genetics is to some extent compensated for by the declining costs of next-generation DNA sequencing, which have made it affordable to identify resistance mutations by whole genome sequencing or transcriptome sequencing of drug-resistant mutants [16,17]. In addition, reverse genetic tools were developed that have helped to overcome the lack of forward genetics.

Inducible RNA interference (RNAi) libraries were used as a high-throughput method for genome-wide loss-of-function studies in *T. brucei* [18,19]. Performing RNAi induction followed by drug selection confirmed the role of the known drug resistance genes *TbAT1*, *AAT6* and *NTR1* as determinants of susceptibility to melarsoprol, eflornithine and nifurtimox, respectively [20,21]. An experimental trypanocide whose target was validated by RNAi is 4-[5-(4-phenoxyphenyl)-2H-pyrazol-3-yl]morpholine, a molecule that turned out to function as a hyperactivator of *T. brucei* adenosine kinase [22,23]. These approaches obviously require the presence of a functional RNAi system in the target cell (Table 1). This, however, is not the case for *T. cruzi* and most species of *Leishmania*, with the notable exception of *L. braziliensis* [9]. Parsimony suggests that the common trypanosomatid ancestor had been competent of RNAi and that the genes for Argonaute and Dicer proteins were lost multiple times in the subsequent course of evolution [24].

Genome editing by CRISPR-Cas9 is more generally applicable. First established for *T. cruzi* [12,13,25], it was also successfully applied to *Leishmania* [14,26] and *T. brucei* [11,27,28]. Further improvements simplified the genetic manipulation of the trypanosomatids, providing a high-throughput system for large-scale genetic knock-out screens [11,29,30]. Another high-throughput tool that has helped to understand the molecular genetics of drug action in *Leishmania* are cosmid libraries. Cos-Seq is based on the selection for enriched loci under drug pressure with subsequent sequencing and candidate gene identification [31,32]. Targeted overexpression of candidate genes is possible as well; with DNA repair genes, this has provided new insights into the mode of action of benznidazole in *T. cruzi* [33]. Inducible gene overexpression libraries have been developed for trypanosomatids that allow identifying drug targets by screening for genes whose overexpression causes drug resistance [34].

Proteomic techniques are applicable to trypanosomatid parasites as well [35,36]. Chemical proteomics combine chemistry to synthesize a drug-derived probe with biology to search for the target protein. This is achieved either by affinity chromatography, in which the chemical probe is immobilized on a matrix and incubated with a cell lysate to fish for proteins, or in situ, where the probe is added to live cells and cross-linked to target proteins e.g., by means of a photoreactive group [37]. The first approach was used to identify MAP kinases and cdc2-related kinases as putative targets of 2,4-diaminopyrimidines in *T. brucei* [38]. The second identified candidate targets of the antiobesity drug orlistat and the antichagasic protease inhibitor K11777 [39,40]. Other innovative techniques for target fishing were developed, such as protein chips, phage display, or the yeast three-hybrid system [41,42], but have to our knowledge not yet been applied to trypanosomatids.

## 3. Artificial Selection for Drug Resistance

Selecting pathogens for drug resistance is a classical experiment in a parasitology laboratory. The first scientist known to have performed it was Paul Ehrlich, the father of chemotherapy. Ehrlich and co-workers infected mice with African trypanosomes and treated the animals with subcurative doses of parafuchsin. They observed that after several passages, the trypanosomes had lost their susceptibility to the drug [43]. Decades later, Alexander Fleming observed the same phenomenon when he cultured bacteria on plates with sublethal concentrations of penicillin [44]. Ehrlich’s main interest in drug resistance was to learn about the nature of the subcellular drug targets. He proposed to use artificially selected drug-resistant pathogens as a ‘therapeutic sieve’ based on cross-resistance profiles [45]. Thus, by testing a new drug candidate against a panel of resistant strains, he was able to tell whether it had a different mode of action. A hundred years later, this concept was incorporated for the development of antimalarials [46]. Today, the selection of drug-resistant mutants in the laboratory mainly serves three purposes: (i) to learn about the mode of drug action, i.e., to identify drug transport pathways and drug targets; (ii) to estimate the risk of emergence of drug resistance in the field based on how quickly resistance evolves in the laboratory; and (iii) to find molecular markers for drug resistance that enable rapid, DNA-based tests. In the following, we focus on the How rather than the Why in the experimental process of laboratory-selection for drug resistance, illustrating different protocols with examples from the ‘TriTryp’ parasites.

### 3.1. Selection In Vitro vs. In Vivo

The selection for drug-resistant trypanosomatids by subcurative dosing of a rodent model of infection is the closest situation to what happens in a treated patient. Therefore, the knowledge that will be gained about the genes and mutations involved in drug resistance is likely to be relevant for the situation in the clinics. A second advantage of selecting in vivo is the high numbers of parasites that can be reached in an infected animal, increasing the probability of success in obtaining a drug-resistant mutant. The main point against in vivo selection is the use of animals per se, if it can be replaced by an in vitro system. In addition, in vivo studies are usually more laborious and expensive than in vitro experiments. A further advantage of in vitro systems is the better control over parameters such as drug concentration and number of parasites.

Overall, there has been a good correlation between the drug resistance phenotypes obtained in vivo and in vitro [47], except in one case, where bloodstream-form *T. b. brucei,* which had been selected in vivo for Cymelarsan resistance, were only weakly resistant to arsenicals in culture [48,49]. A possible explanation is the fast metabolization of melamine-based arsenicals in vivo [50]. Some mechanisms, such as phenomena involving tissue tropism, will only evolve in vivo, whereas others might occur only in vitro. The latter is exemplified by the finding that expression of a particular variant surface glycoprotein (VSG) in *T. brucei* causes suramin resistance [51]. Such a mechanism is hardly sustainable in vivo, where the parasites will be eliminated by the adaptive immune response unless they switch to express another VSG.

The first studies on the selection of trypanosomatids for drug resistance were performed in vivo, as long-term culture systems were unavailable at the time. Early studies with *T. brucei* mainly focused on arsenical resistance [52,53,54,55,56] and on the phenomenon of cross-resistance between melamine-based arsenicals and diamidines [57]. Generally, *T. brucei* spp. were propagated in immunosuppressed mice. The animals were treated with increasing but subcurative concentrations of arsenicals, and the relapsing trypanosomes were passaged to new mice [48,58,59,60]. In a typical experiment, it took eight rounds of infection to obtain a stable resistance phenotype with a resistance factor of 15 [59]. There are only a few studies where *T. cruzi* or *Leishmania* were selected for drug resistance in vivo. Two different approaches were applied, both successfully, to obtain benznidazole-resistant *T. cruzi*. In the first, selection was performed by repeated treatment shocks: an infected mouse at peak parasitemia was treated with a single oral dose of 500 mg/kg benznidazole and after 6 h, the surviving blood trypomastigotes were inoculated into another mouse. This procedure was repeated about 10 days later, again at the peak of parasitemia. After 25 rounds, the obtained *T. cruzi* were unresponsive to benznidazole and cross-resistant to nifurtimox and other nitroimidazoles [61]. The second approach used a lower but constant dose: infected mice were given an oral dose of 100 mg/kg benznidazole daily for 20 consecutive days. Thereafter, the animals were immunosuppressed with cyclophosphamide, and the emerging trypomastigotes in the blood were harvested and inoculated into new mice. Four out of five *T. cruzi* isolates became unresponsive to benznidazole after 2 to 9 rounds of selection [62]. A similar attempt to generate drug-resistant mutants of *L. infantum* and *L. donovani* by repeated subcurative dosing of infected hamsters succeeded remarkably quickly for paromomycin, but not for miltefosine [63]. A different approach, pre-exposing animals before infection, was used to test a possible link between the presence of arsenic in the drinking water and the emergence of antimonial resistance in leishmania. Mice that had been given drinking water with 10 ppm arsenite for 1 month were infected with *L. donovani*. After 28 days, still with arsenite in the animals’ drinking water, the leishmania were passaged to a new group of arsenite pre-exposed mice. After 5 such rounds, the leishmania had become unresponsive to a dose of 500 µg/mL when tested in vitro [64].

There have been many reports on the successful in vitro selection of trypanosomatids for drug resistance (Table 3).

Early in vitro selection experiments were performed even before it was possible to propagate the parasites in culture: African trypanosomes were isolated from an infected rodent at peak parasitemia, exposed in vitro for 1 h to high concentrations of tryparsamide, and reinjected into another animal. After 3 to 13 cycles, the trypanosomes had become less sensitive to tryparsamide [100]. The first culture systems for trypanosomatids were established for the insect stages, i.e., the procyclic, trypomastigote form of *T. brucei* [101], the epimastigote form of *T. cruzi* [102], and the promastigote form of *Leishmania* [103]. When maintained in appropriate medium at 27 °C, these forms readily proliferate in axenic culture and reach densities of over 10^7^ cells per ml before they enter stationary phase. The cultivation of the mammalian stages is less straightforward. Axenic in vitro cultivation is possible for *T. brucei* bloodstream forms [104]. Amastigote *Leishmania*, too, can proliferate in axenic culture if they are kept at low pH to simulate the phagolysosome [105]. Axenic long-term cultivation of *T. cruzi* amastigotes has been reported [106] but is not a standard procedure. Overall, the insect stages of trypanosomatid parasites do not require host cells, are easier to culture than the mammalian stages, and they reach much higher cell densities favoring the selection of drug-resistant mutants. The obvious drawback is that the insect stages are not clinically relevant.

### 3.2. Selection of Insect Stages vs. Mammalian Stages

The key question regarding the choice of the life cycle-stage is whether the mode of drug action is preserved in the insect stages, including drug transport pathway(s) and intracellular target(s). We would argue that if there is no difference in drug susceptibility between mammalian and insect stages, the mode of drug action is likely to be preserved and hence, in vitro selection for drug resistance will be easier and faster with the insect stages even though it is not the stage causing pathogenesis in mammals. However, if the insect stages are clearly less susceptible than the mammalian stages, they might not provide valuable insights on drug resistance (even though the target could be preserved but not essential in the insect stage). Benznidazole for *T. cruzi* is a typical example of a drug that is equally active against either stage, epimastigotes and amastigotes [107] (Table 2). Epimastigote *T. cruzi* selected for benznidazole resistance kept the phenotype after transformation to amastigotes [75]. In contrast, paromomycin resistance that had been selected for in vivo was lost after the amastigote *Leishmania* had been transformed to promastigotes [63]. Regarding *T. brucei*, most drugs used for the treatment of sleeping sickness are much less effective against the procyclic forms in the tsetse fly midgut than against the mammalian bloodstream forms (Table 2). This might be due to the fact that some of the transporters mediating drug uptake are only expressed in the latter [108,109].

An interesting aspect about the relevance of selecting insect stages of trypanosomatid parasites for drug resistance is the question of whether the insect stages ever come into contact with drugs in nature. With African trypanosomes, a scenario that seems plausible is that of an infected tsetse fly that takes a blood meal on a cow that has received nagana drugs, whereupon the procyclic trypanosomes in the fly midgut will be exposed to sublethal drug concentrations. This was experimentally reproduced: *Glossina morsitans* infected with *T. congolense* were fed over 1 month on rabbits that received weekly prophylactic doses of 2 mg/kg Samorin (isometamidium chloride). The flies were then used to infect mice, which in turn served to infect a new group of teneral flies. After four such cycles, the selected *T. congolense* had a significantly lower susceptibility to Samorin than unselected ones, passaged in untreated animals [110].

### 3.3. Selection of a Clone vs. a Population

The probability of obtaining a drug-resistant mutant increases with the genetic diversity of the starting population. This might suggest starting with a heterogeneous population of parasites when selecting for drug resistance in order to speed up the process. However, once the desired drug-resistant mutants have been obtained, their molecular genetic analysis is greatly facilitated if they all derive from the same parental, drug-sensitive clone. Otherwise, there are likely to be too many confounding nucleotide polymorphisms that are unrelated to the resistance phenotype. A typical procedure that facilitates downstream genomics, transcriptomics, and proteomics is to start the selection process with a fresh clone and select several lines independently. Thus, a melarsoprol-resistant mutant of *T. b. rhodesiense* was found to express only two genes (other than *VSG*) at a different level from the parental, melarsoprol-sensitive clone [16], even though it had taken two years of in vitro selection to obtain the mutant [73].

A successful approach towards high-level pentamidine resistance in *T. brucei* was to start with a genetically engineered clone that was already less susceptible to pentamidine since it was homozygously disrupted in the gene *TbAT1*, which encodes an aminopurine permease that also transports diamidine drugs [111]. This starting clone was already 2.4-fold resistant [112]; after several months of in vitro exposure to increasing concentrations of pentamidine, a resistance factor of 130 was obtained [72].

### 3.4. Selection with Mutagens vs. Adaptive Evolution

The use of chemical mutagens poses a similar dilemma as discussed above: it will increase the probability of obtaining a drug-resistant mutant and thus shorten the process of selection. At the same time, it will confront the downstream analyses of the obtained mutants with the challenging task of identifying which of the many mutations are the cause of drug resistance. Since back-crossing to the parental, drug-sensitive line is not feasible with trypanosomatid parasites, resistance selection based on chemical mutagenesis requires a large number of drug-resistant mutants that have been selected in parallel, preferably from the same, freshly cloned parent. Ethyl methanesulfonate (EMS) is frequently used as a mutagen to generate point mutations. It alkylates guanine to ethylguanine, which can form a base pair with thymine, leading to the transition from a G:C pair to A:T. EMS and other mutagens were applied to *L. infantum* promastigotes, followed by plating on media containing either miltefosine or paromomycin, which allowed the isolation of drug-resistant mutants [95]. Finally, the drug itself can be mutagenic. This is the case for ethidium bromide (homidium), which is used in veterinary medicine for *T. congolense* infections. Benznidazole, too, is mutagenic to trypanosomes [79].

### 3.5. Selection under Constant Pressure vs. Sudden Shock

The intuitive approach to select parasites for drug resistance in culture is to apply a steady selective pressure with a sublethal drug concentration, which can be gradually increased as the parasites lose susceptibility. This has been the most commonly used procedure (Table 3). Such an approach is likely to result in the accumulation of several mutations over time, and the phenotype of drug resistance that is ultimately obtained might result from a combination of genetic mechanisms. It is, therefore, imperative to freeze away intermediate samples, which will allow one to determine at what time point a given mutation has occurred. A different, potentially much faster approach is to start with a high inoculum of parasites, expose them to a supposedly lethal concentration of drug, and then wait and see whether, eventually, a population of parasites will recover. This worked well to select *T. brucei* for suramin resistance: when bloodstream-form trypanosomes were incubated with suramin at 5 to 25 fold the IC_50_, all cells seemed to be dead by the following day. However, the cultures were further incubated, and after 6 days a population had regrown that was about 100-fold resistant to suramin [51]. Obviously, such a shock treatment bears the risk that the culture is not going to recover simply because all the parasites are dead. Nevertheless, we think it is a worthwhile experiment to try, since it quickly delivers a (positive or negative) answer.

## 4. Biosafety Considerations and Conclusions

Parasites that have been selected for drug resistance may require additional biosafety measures or even an upgrade in the biosafety level. *T. cruzi* bears the highest biohazard risk among the trypanosomatids. Highly infectious not only by traumatic inoculation but also via mucous membranes, *T. cruzi* is considered in many countries as a pathogen of biosafety level 3 (Table 4). Only two drugs are available for the treatment of an accidental infection with *T. cruzi*, benznidazole and nifurtimox. Both are nitroimidazoles, and they have the same mechanism of action: activation by electron transfer catalyzed by nitroreductase 1 (NTR1), leading to the formation of radicals. Cross-resistance between benznidazole and nifurtimox due to reduced levels of *NTR1* expression is the most frequently observed mechanism [61]. Thus, when *T. cruzi* is being selected for resistance to nitroimidazoles, this will demand even more stringent biosafety measures than required anyhow. This word of caution also applies to selection experiments with the insect stages, because densely grown cultures of epimastigote *T. cruzi* will contain infective metacyclic forms.

In summary, selecting trypanosomatid parasites for drug-resistant mutants requires special care. Moreover, whatever the experimental approach, it also requires patience and luck (in German *Geduld* and *Glück*), two of Paul Ehrlich’s famous Four Gs [113]. With points 3.1 to 3.5, we hope to have provided some guidance about the parameters that need to be considered when planning a drug selection experiment. While there is no bona fide best approach, a methodological consideration of the points outlined above will provide a framework for the successful planning of experiments.

## Figures and Tables

**Table 1 pharmaceuticals-15-00135-t001:** Molecular and cellular characteristics of the three selected trypanosomatids.

	*T. brucei*	*T. cruzi*	*L. donovani*
Genome size [4,5,6]	26.1 Mb	60.4 Mb	32.4 Mb
Protein-coding genes [4,5,6]	9068	~12,000	>8000
Genes of RNAi pathway [7,8,9,10]	present	partially present	absent
RNAi gene silencing [7,8,9,10]	functional	non-functional	non-functional
CRISPR/Cas9 editing [11,12,13,14]	established	established	established
Mammalian stages	extracell. trypomastigotes	intracell. amastigote, extracell. trypomastigote	intracell. amastigote
Vector stages	procyclic trypomastigote, epimastigote, metacyclic trypomastigote	procyclic epimastigote, metacyclic trypomastigote	procyclic promastigote, metacyclic promastigote

**Table 2 pharmaceuticals-15-00135-t002:** Standard drugs and sensitivity of mammalian vs. insect stages. All values are in vitro IC_50_ in µg/ml, original data from our trypanosomatid drug screening platform.

Parasite	Drug	Mammalian Stage Intracellular	Mammalian Stage Axenic	Vector Stage
*T. brucei*	Pentamidine	n.a.	0.001	0.43
	Suramin	n.a.	0.056	>10
	Melarsoprol	n.a.	0.004	0.057
	Eflornithine	n.a.	2.0	>100
	Nifurtimox	n.a.	0.31	1.6
	Fexinidazole	n.a.	0.62	1.2
*T. cruzi*	Benznidazole	0.47	n.a.	3.1
	Nifurtimox	0.14	n.a.	0.87
*L. donovani*	Pentostam	92	220	>1000
	Miltefosine	1.4	0.29	3.8
	Amphotericin B	0.33	0.26	0.03
	Paromomycin	28	>30	10

**Table 3 pharmaceuticals-15-00135-t003:** Reports of successful in vitro selection of trypanosomatids for drug resistance (BSF, bloodstream form; PCF, procyclic form; epi, epimastigotes; pro, promastigotes; trypo, trypomastigotes; RF, resistance factor; n.s., not specified).

Drug	Species	Stage	Mutagen	Pressure	Duration	RF	Ref.
DB75	*T. b. brucei*	BSF	no	steady	2.5 mth	20	[65]
Berenil	*T. b. brucei* Δ*at1*	BSF	no	steady	5 mth	9.2	[66]
Eflornithine	*T. b. brucei*	BSF	no	steady	2 mth	41	[67]
Eflornithine, pentamidine, 1433	*T. b. brucei*	BSF	no	steady	50–120 d	32	[68]
Melarsenoxide cysteamine	*T. b. brucei*	BSF	no	steady	4 mth	15	[59]
Mycophenolic acid	*T. b. gambiense*	PCF	no	steady	n.s.	17	[69]
Nifurtimox	*T. b. brucei*	BSF	no	steady	4.7 mth	8	[70]
Pentamidine	*T. b. brucei*	BSF	no	steady	2 mth	26	[71]
Pentamidine	*T. b. brucei* Δ*at1*	BSF	no	steady	several mth	130	[72]
Pentamidine, melarsoprol	*T. b. rhodesiense*	BSF	no	steady	21 mth	140, 24	[73]
Pyrimidine analogs	*T. b. brucei*	BSF	no	steady	several mth	83–830	[74]
Suramin	*T. b. rhodesiense*	BSF	no	shock	6 d	96	[51]
Benznidazole	*T. cruzi*	epi	no	steady	n.s.	26	[75]
Benznidazole	*T. cruzi*	epi	no	intermittent	15 w	≥4.7	[76]
Benznidazole	*T. cruzi*	epi	no	steady	n.s.	9-26	[77]
Benznidazole	*T. cruzi*	epi	no	steady	several w	n.s.	[78]
Benznidazole	*T. cruzi*	epi	no	steady	4 mth	9–26	[79]
Benznidazole	*T. cruzi*	epi	no	steady	n.s.	23	[80]
Fluconazole	*T. cruzi*	epi	no	steady	4 mth	100	[81]
Nifurtimox	*T. cruzi*	epi	no	steady	8 mth	4	[82]
Nifurtimox	*T. cruzi*	epi, trypo	no	steady	60 d	3–10	[83]
Tubercidin	*T. cruzi*	epi	yes	shock	1 mth	180–260	[84]
CB3717	*L. tropica*	pro	no	steady	3–12 mth	25000	[85]
Allopurinol	*L. infantum*	pro	no	steady	23 w	20	[86]
Amphotericin B, miltefosine paromomycin, Sb^III^	*L. donovani*	pro	no	steady	18 w	11–20	[87]
Methotrexate	*L. tropica*	pro	no	steady	3–11 mth	n.s.	[88]
Arsenite	*L. mex., L. amazon.*	pro	no	steady	1 mth	12	[89]
Hoechst 33342	*L. donovani*	pro	no	steady	n.s.	30	[90]
Daunomycin	*L. tropica*	pro	no	steady	6 mth	62	[91]
Methotrexate	*L. donovani*	pro	no	shock	7–10 gen	n.s.	[92]
Methotrexate	*L. major*	pro	no	steady	n.s.	n.s.	[93]
Miltefosine	*L. donovani*	pro	no	steady	6 mth	15	[94]
Miltefosine, paromomycin	*L. infantum*	pro	yes	steady	10 d	2.5–8.5	[95]
Paromomycin	*L. donovani*	pro	no	steady	3 mth	3	[96]
Pentostam	*L. donovani*	pro	no	steady	n.s.	26	[97]
Primaquin, pentamidine, terbinafine, chloroquine	*L. major*	pro	no	steady	n.s.	2.0–4.4	[98]
Pyrimidine analogs	*L. mex., L. major*	pro	no	steady	12 mth	1–>3500	[99]
Sinefugin	*L. infantum*	pro	no	steady	n.s.	n.s.	[95]
Sodium arsenite	*L. mex., L. amazon.*	pro	no	steady	>1 mth	12	[89]
Sb^III^	*L. major*	pro	no	shock	n.s.	30	[90]

**Table 4 pharmaceuticals-15-00135-t004:** Risk group and biosafety level categorization of *T. brucei*, *T. cruzi* and *L. donovani* (in an infected insect vector, all human-pathogenic trypanosomatids are classified as biosafety level 3).

Countries	*T. brucei*	*T. cruzi*	*L. donovani*
USA	2	2	2
AU/NZ	2	2	n.s.
EU	2 (Tbb), 3 * (Tbr)	3	3 *
UK	2 (Tbb), 3 * (Tbr)	3	3 *
CH	2	3	2

* Limited danger of transmission; usually not transmitted through the respiratory tract (Risk Group Database of the American Biological Safety Association, https://my.absa.org/Riskgroups, accessed on 19 July 2021).

## Data Availability

All data are in the text.

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
