# Peer review of "Laboratory Selection of Trypanosomatid Pathogens for Drug Resistance"

_pharmaceuticals, 2022, doi:10.3390/ph15020135_

Round 1
Reviewer 1 Report
The authors systematically described the selection of three trypanosomatidae for drug resistance. The section 3 provides useful information for researchers who wish to establish drug resistant parasites. Overall, the article is without any clear faults. This review article would be worthy of being accepted in this journal.
Author Response
Thank you very much for the positive comments.
Reviewer 2 Report
Laboratory selection of trypanosomatid pathogens for drug resistance, by Bleistein et al., is a well-written manuscript that concisely summarizes the state of the art regarding different methods to achieve drug resistance in Trypanosoma brucei, T. cruzi, and Leishmania spp. Its publication will be of interest and will help researchers in the field of drug resistance in trypanosomes. There are only minor suggestions aimed at improving the use of English in the table below.
|
Line |
Change |
|
13 |
Replace “lab” with “laboratory” throughout the text (lines 13, 26, 97, 110, etc.), because it is colloquial, shorten spoken word for “laboratory”. |
|
17 |
Add a colon after “resistance”, delete the colon after “trypanosomes”, and modify the wording: “that Paul Ehrlich performed such an experiment for the first time, more than a century ago”. |
|
34-35 |
Regarding trypanosomes it is hardly seen that the mammalian host should be called “obligate intermediate host”, that phrase is usually said for Platyhelminths that have obligate hosts where the adult stage develops. Every host is necessary, but possums for example can have all T. cruzi stages in their urinary glands. Mammalian hosts can become ill due to the infection, in contrast to the insects, but it does not make them “obligate intermediate hosts”. It is more common to read “insect vectors… are the insects that transmit the parasites to the mammalian host…” |
|
87 |
Add “a” before cell lysate. |
|
96-97 |
Change “the” by “a parasitology laboratory”, and “who” by “The first scientist known to have performed it…” |
|
117 |
Rephrase “as close as possible” by “is the closest situation to what happens in a treated patient”. |
Author Response
Thank you very much for the positive comments. We have made all the suggested changes.
Reviewer 3 Report
The manuscript by Beilstein and colleagues on selection of drug resistance in trypanosomatids is a well written, profoundly researched review on this topic that is not only of value for researchers in this field but also for a broader readership.
It can be published after very minor, formal corrections:
- The colloquial “lab” should be replaced by “laboratory” in the text throughout (e.g line 13, l.26 and elsewhere)
- 30: all three species are not only pathogenic to humans, but also animal pathogenic (and thus zoonotic). For drug development and application and the development of resistance the host spectra play an important role and this should not go unmentioned in this article. The authors later refer to this issues, to they may want to add this information to the introduction (“The TriTryp parasites”).
- Table 1: for consistency, in the right column, last line “promastigotes” should be replaced by “promastigote”.
- Table 2: The sources of information for this Table are missing, they should be added as an additional column with reference numbers.
- In some instances, automatic hyphenation produced the wrong syllable division (e.g. line 219-220) or separation of values from units (e.g. line 164-165)
- Line 65: 4-[5-… the name of this chemical is incomplete; “morpholine” is missing after “…3-yl], it seems.
- Table 3: The table is not referred to in the text before, but only in the text on the following page, this should be changed to be compliant with standard rules of table citations; the heading does not describe what can be found in the Table (“selected” here refers to “selected for resistance”, but as a stand-alone Table header it just means “three species selected to write about”. The text in ll. 170-171 describes the contents much better. Lastly, the abbreviations used in the column “stage” should be described in the table header. “n.s.” usually stands for “not significant”, so for clarity this abbreviation should also be explained in the header.
Author Response
Thank you very much for the positive comments. We have made all the suggested changes and corrections (except for turning off the auto-hyphenation feature in the MDPI template; this is something we would rather leave to the typesetter of MDPI).